Modeling a deep transfer learning framework for the classification of COVID-19 radiology dataset

http://orcid.org/0000-0002-9833-4951 Fayemiwo Michael Adebisi 1
http://orcid.org/0000-0001-8505-1816 Olowookere Toluwase Ayobami 1
http://orcid.org/0000-0003-1266-0535 Arekete Samson Afolabi 1
http://orcid.org/0000-0002-7913-3281 Ogunde Adewale Opeoluwa 1 ogundea@run.edu.ng
http://orcid.org/0000-0002-1476-3610 Odim Mba Obasi 1
Oguntunde Bosede Oyenike 1
Olaniyan Oluwabunmi Omobolanle 1
Ojewumi Theresa Omolayo 1
Oyetade Idowu Sunday 1
http://orcid.org/0000-0002-4207-3756 Aremu Ademola Adegoke 2
http://orcid.org/0000-0001-5440-5546 Kayode Aderonke Anthonia 1
1 Department of Computer Science, Redeemer’s University , Ede, Osun , Nigeria
2 Radiology Department, Ladoke Akintola University of Technology , Ogbomoso, Oyo , Nigeria
Chicco Davide
Electronic publication date: 2021 Aug 3
Publication date: 2021
Volume: 7
Electronic Location ID: e614
Received 2021 Mar 12; Accepted 2021 Jun 7
Copyright: © 2021 Fayemiwo et al.
Copyright year: 2021
Copyright holder: Fayemiwo et al.
License: This is an open access article distributed under the terms of the Creative Commons Attribution License, which permits unrestricted use, distribution, reproduction and adaptation in any medium and for any purpose provided that it is properly attributed. For attribution, the original author(s), title, publication source (PeerJ Computer Science) and either DOI or URL of the article must be cited.
License URL: https://creativecommons.org/licenses/by/4.0/

Keywords: Convolutional neural networks, Coronavirus, COVID-19 test results, Deep transfer learning, Machine learning, VGG-16, VGG-19

Funding: The authors received no funding for this work.

==============================
Severe Acute Respiratory Syndrome Coronavirus 2 (SARS-Coronavirus-2 or SARS-CoV-2), which came into existence in 2019, is a viral pandemic that caused coronavirus disease 2019 (COVID-19) illnesses and death. Research showed that relentless efforts had been made to improve key performance indicators for detection, isolation, and early treatment. This paper used Deep Transfer Learning Model (DTL) for the classification of a real-life COVID-19 dataset of chest X-ray images in both binary (COVID-19 or Normal) and three-class (COVID-19, Viral-Pneumonia or Normal) classification scenarios. Four experiments were performed where fine-tuned VGG-16 and VGG-19 Convolutional Neural Networks (CNNs) with DTL were trained on both binary and three-class datasets that contain X-ray images. The system was trained with an X-ray image dataset for the detection of COVID-19. The fine-tuned VGG-16 and VGG-19 DTL were modelled by employing a batch size of 10 in 40 epochs, Adam optimizer for weight updates, and categorical cross-entropy loss function. The results showed that the fine-tuned VGG-16 and VGG-19 models produced an accuracy of 99.23% and 98.00%, respectively, in the binary task. In contrast, in the multiclass (three-class) task, the fine-tuned VGG-16 and VGG-19 DTL models produced an accuracy of 93.85% and 92.92%, respectively. Moreover, the fine-tuned VGG-16 and VGG-19 models have MCC of 0.98 and 0.96 respectively in the binary classification, and 0.91 and 0.89 for multiclass classification. These results showed strong positive correlations between the models’ predictions and the true labels. In the two classification tasks (binary and three-class), it was observed that the fine-tuned VGG-16 DTL model had stronger positive correlations in the MCC metric than the fine-tuned VGG-19 DTL model. The VGG-16 DTL model has a Kappa value of 0.98 as against 0.96 for the VGG-19 DTL model in the binary classification task, while in the three-class classification problem, the VGG-16 DTL model has a Kappa value of 0.91 as against 0.89 for the VGG-19 DTL model. This result is in agreement with the trend observed in the MCC metric. Hence, it was discovered that the VGG-16 based DTL model classified COVID-19 better than the VGG-19 based DTL model. Using the best performing fine-tuned VGG-16 DTL model, tests were carried out on 470 unlabeled image dataset, which was not used in the model training and validation processes. The test accuracy obtained for the model was 98%. The proposed models provided accurate diagnostics for both the binary and multiclass classifications, outperforming other existing models in the literature in terms of accuracy, as shown in this work.

Introduction

Viral pandemics are usually a serious threat to the world, and coronavirus disease 2019 (COVID-19) is not an exception. According to a COVID-19 report of the World Health Organization (WHO), coronaviruses are from a large family of viruses that cause illness in animals or humans (WHO, 2020). Numerous coronaviruses have been reported as the cause of respiratory disease in humans, ranging from the common cold to more serious illnesses like Middle East Respiratory Syndrome (MERS) and Severe Acute Respiratory Syndrome (SARS). This newly discovered coronavirus has caused the 2019-novel coronavirus (COVID-19) disease. COVID-19 was initially observed in the Wuhan province of China and has spread to all parts of the world (Nadeem, 2020). COVID-19 was recognized as a contributory virus by Chinese authorities on January 7, 2020. The Director-General of WHO, on the January 30, 2020 reported that the epidemic constitutes a Public Health Emergency of International Concern (PHEIC), based on the recommendations made by the Emergency Committee. WHO activated the R&D Blueprint in reaction to the occurrence to speed up diagnostics, vaccines, and therapeutics for this new coronavirus (WHO, 2020). The International Committee on Taxonomy of Viruses named the novel coronavirus “severe acute respiratory syndrome coronavirus 2 (SARS-Coronavirus-2 or SARS-CoV-2)”. Globally, as of March 10, 2021, there have been 117,332,262 confirmed cases of COVID-19, including 2,605,356 deaths, reported to WHO. Also, as of March 9, 2021, a total of 268,205,245 vaccine doses have been administered (WHO, 2021). Specifically, in Nigeria, the Nigeria Centre for Disease Control (NCDC) report on March 11, 2021, showed 394 new confirmed cases recorded. To date, 159,646 cases have been confirmed, 139,983 patients have recovered and discharged, and 1,993 deaths have been recorded in 36 states and the Federal Capital Territory (Nigeria Centre for Disease Control, 2021).

Numerous researchers globally are putting their efforts together on collecting data and developing solutions. The persistent focus has been on advancing key performance indicators, for example, continually enhancing the speed of case detection, segregation, and early cure. The execution of these containment procedures has been sustained and enabled by the pioneering and aggressive use of cutting-edge technologies. Measures such as immediate case detection and isolation, rigorous close contact tracing and monitoring/quarantine, and direct population/community engagement have been considered in lessening COVID-19 illness and death. This work, therefore, is aimed at using Artificial Intelligence (AI), specifically machine learning, to identify those who are at risk of contracting COVID-19 to aid early diagnosis. According to British Broadcasting Corporation (BBC) (2020), a superhuman attempt is required to ease the deaths due to the global epidemic. AI may have been overestimated—but in the case of medicine, it already has established evidence. According to Arora et al. (2020), the role of AI is going to be crucial for predicting the outcome based on symptoms, CT-Scan, X-ray reports, etc.

Laboratory checking of suspected cases is characterized by extended periods of testing and an exponential rise in test requests (Kobia & Gitaka, 2020). Quick diagnostic tests with shorter turnaround times of between 10 and 30 minutes have been developed to ease the problem. However, many are presently going through clinical validation and is not in regular use (ECDC, 2020). While awaiting results, there is a need to continue to self-isolate. Once results are received, there is a need to remain on self-isolation until the symptoms resolve after being in seclusion for at least 14 days. If the symptoms worsen during the seclusion time or continued after 14 days, the patient has to contact the accredited healthcare providers. Rapid Test Kits can even deliver results after hours.

Pneumonia has been described as the most severe and frequent manifestation of COVID-19 infection (Huang et al., 2020); therefore, chest imaging including readily available and affordable chest radiograph (X-rays) remains an essential factor in the diagnosis and evaluation of COVID-19 patients (Rubin et al., 2020). However, the availability of radiologists to report the chest images is another obstacle. Therefore, there is a need to develop computer algorithms and methods to optimize screening and early detection; that is the primary purpose of this research in which deep learning, most especially Convolutional Neural Network (CNN), is deployed. Deep learning provides the chance to increase the accuracy of the early discovery by automating the primary diagnosis of medical scans (Madan, Panchal & Chavan, 2019). The CNN belongs to a category of Deep Neural Networks (DNN), which comprise several layers that are hidden like convolutional layers. The convolutional layers come with the non-linear activation function, a rectified linear unit (ReLU layer), a pooling layer, and a fully connected normalized layer. CNN divides weights in the convolutional layer, thereby decreasing the memory footprint and increasing the performance of the network (Sasikala, Bharathi & Sowmiya, 2018). The objective of this paper is to classify a real-life COVID-19 dataset consisting of X-ray images using a novel Deep Learning Convolutional Neural Network Model. Data from chest X-rays were used because most hospitals have X-ray machines, and the COVID-19 X-ray dataset is now available on the web. The remaining parts of this paper are organized as follows: literature review, materials and methods, experimentation and results, evaluation of results, conclusion and future works.

Literature review

COVID-19 is predominantly a respiratory illness, and pulmonary appearances constitute the main presentation of the disease. SARS-CoV-2 infects the respiratory system but may also affect other organs, as reported in some studies. Renal dysfunction (Chu et al., 2005; Xu et al., 2020), gastrointestinal complications (Pan et al., 2020), liver dysfunction (Huang et al., 2020), cardiac manifestations (Zhou et al., 2020), mediastinal findings (Valette, du Cheyron & Goursaud, 2020), neurological abnormalities, and haematological manifestations (Song & Shin, 2020) are among the reported extrapulmonary features. Some of the clinical symptoms of COVID-19 are cough, expectoration, asthenia, dyspnoea, muscle soreness, dry throat, pharyngeal dryness and pharyngalgia, fever, poor appetite, shortness of breath, nausea, vomiting, nasal obstruction and rhinorrhoea. A study on COVID-19 credited to WHO, stated that the disease does not exhibit distinct symptoms, and patients’ symptoms can vary from fully asymptomatic to extreme pneumonia and death (WHO, 2020). Nevertheless, certain symptoms, such as dry cough, fever, dyspnea and fatigue, were confirmed to be more prevalent in COVID-19 patients. Sore throat, nasal inflammation, fever, arthralgia, chills, diarrhoea, hemoptysis, nausea and conjunctival congestion are some of the other clinical symptoms (Behzad et al., 2020). Other non-specific symptoms include loss of smell and taste, dermatologic eruptions, delirium, and a general decline in health (Recalcati, 2020). Because of the wide range of clinical indications and the growing global burden of COVID-19, it is critical to promptly scale up diagnostic ability to diagnose the virus and its risks.

Reverse transcription-polymerase chain reaction (RT-PCR) is the primary clinical instrument currently in use to detect COVID-19. It uses respiratory specimens for testing (Wang et al., 2020a). RT-PCR is used as a reference method for detecting COVID-19 patients; however, the technique is expensive, manual, complicated, time-consuming, and requires specialized medical personnel. Alternatively, X-ray imaging is an easily accessible tool that can be excellent in the COVID-19 diagnosis. Chest imaging is an essential part of evaluating respiratory complications, which remain one of the most familiar presentations ranging from acute respiratory distress syndrome to respiratory failure (Huang et al., 2020). Chest imaging has been defined as an efficient screening method for detecting pneumonia, with a sensitivity of 97.5 percent for COVID-19 (Nabila et al., 2020; National Health Commission of People’s Republic of China, 2020). Provided COVID-19's preference for the respiratory system, chest radiography (X-ray), CT of the thorax, and/or Ultrasound have been verified not only as case management and screening methods for COVID-19, but also as a way of reducing infection transmission through early detection in initially False-negative RT-PCR tests (Rubin et al., 2020). Imaging tests are useful for generating clinically actionable outcomes, which can be used to determine a diagnosis or to guide management, triage, or treatment. Costs such as the risk of exposure to radiation to the patient, the risk of COVID-19 transmission to uninfected health care staff and other patients, the use of personal protective equipment (PPE), and the need for sanitation and interruption of radiology rooms in resource-constrained settings reduce the benefit (Kooraki et al., 2020).

The role of imaging includes the detection of early parenchymal lung disease, disease progression, complications, and alternative diagnoses, including acute heart failure from COVID-19 myocardial injury and pulmonary thromboembolism (Driggin et al., 2020). Although CT is more sensitive than chest radiography, chest radiography remains the first-line imaging modality in COVID-19 patients because of its availability, affordability, reduced radiation risk, and ease of decontamination (Fatima et al., 2020). Hence, this research work utilized chest radiographs (X-ray) for identifying COVID-19 in patients. Bilateral, peripheral, lower zone prevalent ground-glass opacities are common chest radiograph observations in COVID-19 patients (Vancheri et al., 2020). Other possible findings include normal, unilateral, or bilateral reticular alterations, consolidations, ground-glass opacities and pleural effusion (Hamid, Mir & Rohela, 2020).

COVID-19 is transmitted through droplets from coughing or sneezing and on close contacts with infected persons. Propagation of based on infected surfaces are considered as the major means of transmitting SARS-CoV-2. However, patients going through screening are protected and scanned through the use of treated tools (Kooraki et al., 2020). The incubation period of COVID-19 is usually about 14 days, during which it attacks the lung. Different countries recommend personal protection equipment (PPE). According to the Centers for Disease Control and Prevention, radiology personnel should use a face mask, glasses or face shield, sleeves, and an isolation garment. A surgical cap and foot covers are required in countries with more rigorous PPE guidelines, whereas a surgical mask, goggles, or face shield are recommended in countries with less strict PPE guidelines (Centres for Disease Control & Prevention, 2020). COVID-19 is highly contagious, and the symptoms begin to appear 5–6 days after contracting it either from the droplet or close contact with an infected person. According to Wang, Tang & Wei (2020), the period between the manifestation of symptom and demise ranges from 6–41 days, depending on the age and immune system of the patient. The period is shorter among older people (Bai et al., 2020; Hamid, Mir & Rohela, 2020).

COVID-19 shows some unique clinical symptoms like targeting the lower airway that manifests in sore throat, sneezing, and rhinorrhoea. In addition, the chest radiographs’ result in some cases possess “infiltrate in the upper lobe of the lung” related to growing dyspnea with hypoxemia (Hamid, Mir & Rohela, 2020). COVID-19 has become endemic and rapid diagnosis is imperative to identify patients and carriers for possible isolation and treatment to curb the spread of the disease. Attempts have been made to diagnose the disease, but many are slow and not accurate in that they often give false-negative and false-positive results. This section examines some of the work done in this area.

Over comprehensive detection of each aggressive spread chain, Lokuge et al. (2020) proposed an accurate, fast, and flexible tracking approach for spotting all residual COVID-19 group transmission. Using surveillance evaluation methods, they considered efficiency and sensitivity in the classification of population transmission chains by testing primary care fever and cough patients, hospital cases, or asymptomatic community members. They also varied the number of duplications, monitoring capacities, and the prevalence of COVID-19 and non-COVID-19 fever and cough. The findings of the study revealed that testing both syndromic fever and cough primary care presentations, as well as precise and diligent case and touch monitoring and assessment, allows for proper primary direct detection and elimination of COVID-19 population transmission. Even with optimized test sensitivity, measures such as combining these approaches could allow for increased case discovery if testing capacity is minimal. The impact analysis of movement restriction as a result of emergence of COVID-19 was carried out by Hyafil & Moriña (2020). The study looked at the impact of the steps put in place in Spain to combat the epidemic. The instances and the influence of the imposed restriction to movement on the multiplicative quantity of hospitalization reports were estimated. The projected figure of instances displayed a rapid rise towards total movement restriction as imposed. The primary replication rate reduced meaningfully from 5.89 (95% CI [5.46–7.09]) before the lockdown to 0.48 (95% CI [0.15–1.17]) after the lockdown. The study found that managing a pandemic in the magnitude of COVID-19 was very intricate and required timely decisions. The significant modifications found in the infestation rate displayed that employing inclusive participation in the first phase was vital in reducing the effect of a possible transferrable threat. The paper likewise stressed the significance of dependable up-to-date epidemiological facts to precisely measure the influence of Public Health guidelines on the virus-related outburst.

Yu et al. (2020) observed the mounting evidence that suggested that there remained a hidden collection of COVID-19 asymptomatic but transferable cases and that approximating the count of disease instances without symptoms was essential in knowing the virus and curtailing its transmission; though, it was reported that it was difficult to precisely calculate the spread of the infection. A machine learning-based fine-grained simulator (MLSim) was proposed to combine many practical reasons, such as disease development during the maturation phase, cross-region population movement, unobserved patients without symptoms, preventative measures and confinement resilience, to estimate the number of asymptomatic infection, which is critical in understanding the virus and accurately containing its spread. Digital transmission mechanisms with many unspecified variables were used to simulate the relationships between the variables, which were calculated from epidemic data using machine learning approach. When MLSim learned to closely compare and contrast real-world data, it was able to simulate the instances of patients without symptoms as well. The accessible Chinese global epidemic data helped the MLSim to train better. The analysis indicated that fine-grained machine learning simulators could improve the modelling of dynamic real-world infection transmission mechanism, which can aid in the development of balanced mitigation steps. The simulator equally showed the possibility of a great amount of undiscovered disease risk, posing a major threat to containing the virus. COVID-19 was modelled using a composite stochastic and deterministic concept that allowed for time-varying transmission capacity and discovery chances (Romero-Severson et al., 2020). Iterative particle sorting was used to adapt the model to a historical data study of occurrence and casualty figures from fifty-one countries. The report confirmed the fact that the spread rate is decreasing in forty-two of the fifty-one countries surveyed. Out of the forty-two countries, 34 showed a big significant proof for subcritical transmission rates, though the turndown in novel cases was moderately slow in comparison to early development rates. The study concluded that attempts to reduce the occurrence of COVID-19 by social distancing were successful. They could, however, be improved and retained in various regions to prevent the disease from resurfacing. The study also proposed other approaches to manage the virus before the relaxation of social distancing efforts.

The challenges associated with the storage and security of COVID-19 patients’ data were highlighted by ElDahshan, AlHabshy & Abutaleb (2020). The authors pointed out that the variety, volume, and variability of COVID-19 patients data required storage in NoSQL database management systems (NoSQL DBMS). It was noted that available NoSQL DBMSs were fraught with security challenges that rendered them unsuitable for storing confidential patient data. Academic institutions, research centres, and enthusiasts find it difficult to select the most suitable NoSQL DBMS because there are myriads of them without standard ways of determining the best. Thus, the study presented an inventive approach to selecting and securing NoSQL DBMS for medical information. The authors outlined the five most common NoSQL database groups, as well as the most common NoSQL DBMS forms affiliated with every one of them. In addition, their research included a comparison of the various types of NoSQL DBMS. The paper provided an efficient solution to myriads of security challenges, ranging from authorization, authentication, encryption, and auditing in storing and securing medical information utilizing a collection of web service-based functions.

Guerrero, Brito & Cornejo (2020) used a mathematical model to depict a sneezing individual in an urban setting with a meteorological wind of medium strength. The proliferation of airborne route was demonstrated using a Lagrangian method and a wall-modeled Large Eddy Numerical simulation. The results showed that the dimensions of two kinds of droplets differ in size: larger droplets (400–900 μm) scatter between 2–5 m in 2.3 s, whereas smaller (100–200 μm) droplets are transported in a larger, more impressive array between 8–11 m by the windy conditions in 14.1 s on average. Knowing the ambiguity of possible infection in this way aids in the development of solutions for the possibility of adopting tougher self-care and distance policies.

To distinguish communicable acute abdomen patients suspected of COVID-19, Zhao et al. (2020) proposed a forecasting model identified as a monogram and scale. The analytical framework was built on the basis of a retrospective case study. In a training cohort, the model was formulated using LASSO regression and multivariable logistic regression method. In the training and testing cohorts, standard curve evaluated the efficiency of the monogram, receiver operating characteristic (ROC) curves, decision curve analysis (DCA), and clinical effect curves. According to the monogram, a simpler testing scale and management algorithm was developed. In the testing cohort, the CIAAD monogram demonstrated strong differentiation and standardization, which was approved. The CIAAD monogram was clinically useful, according to decision curve research. The monogram was condensed even further into the CIAAD standard. The estimated Bayesian Computation method was utilized by Vasilarou et al. (2020) to determine the parameters of a demographic scenario engaging an exponential growth of the size of the COVID-19 populations and revealed that rapid exponential development in population size could sustain the monitored polymorphism patterns in COVID-19 genomes.

Amrane et al. (2020) adopted a genetic approach using a rapid virological diagnosis on sputum and nasopharyngeal samples from suspect patients. Two real-time RT-PCR systems employing a “hydrolysis probe and the LightCycler Multiplex RNA Virus Master Kit” were used. The primary technique probes the envelope protein (E)-encoding gene and used a synthetic RNA positive control. The subsequent system targeted the spike protein-encoding gene (forward primer, reverse primer, and probe) and used synthetic RNA positive control methods. Bai et al. (2020) proposed the use of medical technology through the internet of things (IoT) to develop an intelligent analysis and treatment assistance programme (nCapp). The conceptual cloud-based IoT platform includes the basic IoT functions as well as a graphics processing unit (GPU). To aid in deep mining and intelligent analysis, cloud computing systems were linked to existing electronic health records, image cataloguing, picture cataloguing, and interaction. Li et al. (2020) examined chest images for the analysis of COVID-19. High-resolution Computed tomography (HRCT) was implemented for analysis of the virus infection. CT scans were taken with the following parameters: 120 KV; 100–250 mAs; collimation of 5 mm; the pitch of 1–1.5; and 512 × 512 matrix. The images were reconstructed by high resolution and conventional algorithms. The experiments were repeated several times, running into days for each patient. The High-resolution CT objectively evaluated lung lesions giving a better understanding of the pathogenesis of the disease.

Long et al. (2020) evaluated the suitability of Computed Tomography (CT) and real-time reverse-transcriptase-polymerase Chain Reaction (rRT-PCR). A clinical experiment with life data was executed, and the results presented showed that CT examination outperformed that of rRT-PCR at 97.2% and 84.6%, respectively. In Vaishya et al. (2020), seven critical AI applications for the novel COVID-19 were recognized to perform vital roles in screening, analyzing, tracking, and predicting patients. Application areas identified comprised early detection and diagnosis of the infection, treatment monitoring, individuals contact tracing, projection of cases and mortality, drugs and vaccines development, lessening healthcare workers’ assignment, and deterrence of the disease by providing updated supportive information.

Zhang et al. (2020) conducted a survey that presented some proofs of mental distress and associated predictors amongst adults in the current COVID-19 pandemic in Brazil. The data was composed of 638 adults from March 25 to 28, 2020, about one month after the index case was confirmed in São Paulo. Female adults, who were young, more trained, and practiced less, recorded higher levels of distress, with 52 percent experiencing mild-to-moderate distress and 18.8 percent suffering extreme distress. The study’s findings also revealed that a person’s distance from Sao Paulo, the epicenter, had a direct connection with the psychological distress they were experiencing. For the older population who worked the least, the “typhoon eye effect” was more potent. Adults who lived far off the worst hit geographic area and did not go to work in the week preceding the survey were the most vulnerable. The paper concluded that recognizing the predictors of suffering would allow mental health services to improve target finding and assist the more mentally defenseless adults in the crisis. Ghafari et al. (2020) assessed the challenges and indications of concern of the COVID-19 in Iran. The heterogeneous COVID-19 casualty levels around the country in fourteen university hospitals in Tehran were investigated, and it was revealed that the recorded cases on 13/03/2020 indicated just under 10% of symptomatic patients in the population in adolescents. The finding indicated that there was a major inaccurate reporting of cases in Iran. The study suggested that strict measures be implemented throughout a period of widespread underreporting in order to prevent the healthcare system from being exhausted within a month. Further studies on efficient diagnosis, detection, and vaccine of the virus are continuing for two main reasons: partly because the disease is new, and secondly because available research efforts have not been able to address the concerns effectively. Therefore, in this paper, a deep learning modeling framework for efficient identification, classification, and provision of new insights for the diagnosis of COVID-19 was presented. Also, the prediction of probable patients of the novel COVID-19 using radiology scanned images of suspected patients were shown.

Zivkovic et al. (2021) proposed a hybrid comprising machine learning Adaptive Neuro-Fuzzy Inference System (ANFIS) and enhanced Bio-inspired Beetle Antennae Search (BAS) Algorithm referred to as CESBAS-ANFIS. The hybrid study was carried out with a view to improving the existing time-series prediction algorithms for forecasting COVID-19 new cases. An improved BAS algorithm was adopted to update the parameters of ANFIS. A prediction model for the virus outbreak was formulated using ANFIS trained by the improved BAS algorithm to enhance the prediction accuracy of new cases of COVID-19. CESBAS was introduced to update ANFIS parameters, thereby solving parameters’ optimization problem of machine learning techniques for prediction. Cauchy mutation operator was incorporated into the original BAS called CESBAS (Cauchy Exploration Strategy BAS) to improve exploration ability and solution diversity deficiencies observed. The proposed method consists of five layers; one input layer, two hidden layers, one layer for conclusion parameters and the output layer that presented the forecasted value. The proposed model with other hybrid models was tested under the same conditions on two datasets; one dataset from WHO and the second from the “our world in data” website, and their results were compared. CESBAS-ANFIS showed superior performance compared to other hybrid techniques such as ABC-ANFIS, BAS-ANFIS and FPA-ANFIS.

Irfan et al. (2021) explored the contributions of hybrid deep neural networks (HDNNs), chest X-rays and computed tomography (CT) in the detection of COVID-19. The work employed X-ray imaging and CT to develop the HDNNs for predicting the early infection of COVID-19. The HDNNs were trained and tested on five thousand (5,000) images collected from five different sources (public and open), comprising 57% males and 32% females, and 3,500 infected and 1,500 healthy controls within an age group of 38–55 years. The proportion of the test dataset to the training dataset was 20:80, and classification accuracy of 99% was achieved with the HDNNs on the test dataset. The results of the performed experiments showed that the new multi-model and multi-data approach achieved improved performance over the traditional machine learning models.

Elzeki et al. (2021) presented a new deep learning computer-aided scheme for rapid and seamless classification of COVID-19. Consisting of three separate COVID-19 X-ray datasets, the study presented the COVID Network (CXRVN) model for assessing grayscale chest X-ray images. The scheme was implemented on three different datasets using MatLab 2019b. A comparison was made with the pre-trained models of AlexNet, GoogleNet and ResNet using the mini-batch gradient descent and Adam optimizer to aid the learning process. Performance evaluation results of the model using F1 score, recall, sensitivity, accuracy, and precision, with generative adversarial network (GAN) data augmentation revealed that the accuracy for the two-class classification was 96.7%. In comparison, the three-class classification model reached 93.07%. However, the authors pointed out that increased availability of datasets could improve the performance of future methodologies. In addition, it was stated that the model could be enhanced by employing computed tomography (CT-images) and studying different updated cases of the COVID-19 X-ray images.

From the reviews presented in this section, it was observed that some of the existing works reported low accuracies, used imbalanced datasets, and there was no evidence of the use of some standard evaluation metrics such as Matthew’s Correlation Coefficient and Cohen’s Kappa Statistics in their approaches. The aforementioned issues were addressed in this paper. In addition, the performance of the VGG-16 and VGG-19 networks, the forms of the VGGNet Architecture, in predicting COVID-19 X-ray datatsets were compared in this study.

The VGGNet architecture

The VGGNet, proposed by Simonyan & Zisserman (2015), is a convolutional neural network that performed very well in the ImageNet Large Scale Visual Recognition Challenge (ILSVRC) in 2014. The VGG-16 and VGG-19 networks are forms of the VGGNet Architecture. The networks accept color images with size 224 × 224 and three channels (Red, Green and Blue) as their input data. The images pass through convolutional layers that are stacked on top of each other, in which there is a limited reactive field of 3 × 3 and stride of 1 in the convolutional filter. The convolutional kernel employs row and column padding such that the resolution before the convolution is retained after the processing of the images. Max-pooling is then done over a max pool window of size 2 × 2 with a stride of 2 (Simonyan & Zisserman, 2015).

The network of the VGG-16 CNN has 13 convolutional layers (that is, 3 × 3 convolutional layers in blocks that are stacked on top of one another with growing depth). Two blocks house two 3 × 3 convolutional layers of the same setup in a sequential arrangement, while three blocks have three 3 × 3 convolutional layers of the same configuration in a sequential arrangement. As such, the VGG-16 has two contiguous blocks of two convolutional layers, with each block accompanied with a max-pooling. Also, it has three continuous blocks of three convolutional layers, with each block accompanied with a max-pooling. In all, there are five max-pooling layers in the architecture. Max pooling layer handles the reduction of the volume size after each block that contains two convolutional layers and after each block that contains three convolutional layers. The informative features are obtained by these max-pooling layers that are applied at the earlier specified stages in the network. The VGG-16 further has two fully-connected layers, each with 4,096 nodes and one fully-connected layer with 1,000 nodes, one node each for each of the 1,000 categories of images in the ImageNet database on which the network was pre-trained, and is followed by the SoftMax classifier (Simonyan & Zisserman, 2015), as presented in the VGG Architecture, which can be found in Frossard (2016).

The network of the VGG-19 CNN has 16 convolutional layers (that is, 3 × 3 convolutional layers in blocks that are stacked on top of one another with growing depth). Two blocks house two 3 × 3 convolutional layers of the same setup in a sequential arrangement, while three blocks have four 3 × 3 convolutional layers of the same configuration in a sequential arrangement. In other words, the VGG-19 has two continuous blocks of two convolutional layers, with each block accompanied by a max-pooling. It also has three contiguous blocks of four convolutional layers, with each block accompanied with a max-pooling. In all, there are five max-pooling layers in the architecture. The Max-pooling layer handles the reduction of the volume size after each block that contains two convolutional layers and after each block that contains four convolutional layers. The informative features are obtained by these max-pooling layers that are applied at the earlier specified stages in the network. The VGG-19 further has two fully-connected layers, each with 4,096 nodes and one fully-connected layer with 1,000 nodes, one node each for each of the 1,000 categories of images in the ImageNet database on which the network was pre-trained, and is followed by the SoftMax classifier (Simonyan & Zisserman, 2015).

Materials & methods

This study focused on diagnosing the COVID-19 chest X-ray dataset using a deep learning convolutional neural network (CNN). The CNN comprises one or more convolution layers and then followed by one or more fully connected layers as obtained in a standard multilayer neural network. COVID-19 Radiology Dataset (chest X-ray) for Annotation and Collaboration was collected from the Kaggle website (a database collated by researchers from Qatar University and the University of Dhaka) with collaborators from Pakistan and Malaysia, and some medical doctors, and Mendeley dataset repository. The data was preprocessed, and the median filter was employed to restore the image undergoing evaluation by mitigating the severity of collection degradations. Manikandarajan & Sasikala (2013) mentioned some preprocessing and segmentation strategies that were used. Each data point was replaced by the average value of its neighbors, and itself, in the median filter. As a result, data points that differed significantly to their neighbors were removed. Following the preprocessing of the image dataset, the images were sectioned by using a simulated annealing algorithm. Feature extraction and classification were done using CNN. The neural network-based convolutional segmentation was implemented in Jupyter Notebook using Python programming language and the model was built using sample datasets for the system to recognize and classify the COVID-19. The model generated can be used to develop a simple web-based application that medical personnel handling COVID-19 tests could use to input new cases and quickly predict the presence of the COVID-19, with a very high level of accuracy.

Dataset description and preprocessing

Chest X-ray images were selected from a repository of COVID-19 positive cases’ chest X-ray images, as well as regular and viral pneumonia images, which were collated by researchers from Qatar University and the University of Dhaka, along with collaborators from Pakistan and Malaysia, and some medical doctors. There are 219 COVID-19 positive images in their current release, 1,341 normal images, and 1,345 viral pneumonia images (Chowdhury et al., 2020). For multiple representations, the dataset of chest X-ray images for both COVID-19 and normal cases were also selected from the Mendeley dataset repository (El-Shafai & Abd El-Samie, 2020), which contains 5,500 Non-COVID X-ray images and 4,044 COVID-19 X-ray images. This study, therefore, adopted these multisource datasets. Due to limited computing resources, in this study, 1,300 images were selected from each category for model building and validation. In other words, 1,300 images of COVID-19 positive cases, 1,300 normal images, and 1,300 images of viral pneumonia cases, totalling 3,900 images in all. Also, a different set of 470 images (containing 70 COVID-19 images, 200 ViralPneumonia and 200 Normal) were selected and used for testing to obtain an impartial evaluation of a final model. The dataset used in this study can be found at Fayemiwo et al. (2021a). It should be noted here that further descriptions of the datasets were not provided by the authors of the dataset’s sources.

OpenCV (Bradski & Kaehler, 2008) was used for loading and preprocessing images in the dataset. Each image was loaded and preprocessed by performing a conversion to RGB channel and changing the size of the images to 224 × 224 pixels to be ready for the Convolutional Neural Network. Pixel intensities were then scaled to the range [0, 1] and converted the data and labels to NumPy array format. Labels were then encoded using a one-hot encoder while training/testing splits were created. To ensure that the model generalizes well, data augmentation was performed by setting the random image rotation to 15 degrees, random range zooming to 0.15, random shift of width and height to 0.2, random shear range to 0.15, randomly flipping half of the images horizontally by setting horizontal_flip = True and fill_mode to nearest. One thousand, nine hundred and fifty (1,950) images were allocated initially to train the model in the binary classification task, resulting in 78,000 training images after augmentation. Two thousand, nine hundred and twenty-five (2,925) images were allocated initially for training the model in the multiclass classification task, which resulted in 117,000 training images after augmentation.

The deep-transfer learning model

This study employed the VGG-16 and VGG-19 Convolutional Neural Networks (CNN) with Deep Transfer Learning (DTL) approach for COVID-19 detection. The DTL approach focused on the storage of weights that have been grown while unraveling some image classification tasks and then engaging them on a related task. Several DTL networks have been proposed, some of which include VGGNet (Simonyan & Zisserman, 2015), GoogleNet (Szegedy et al., 2015), ResNet (He et al., 2015), DenseNet (Huang et al., 2017) and Xception (Chollet, 2017). In this paper, VGG-16 CNN and VGG-19 CNN, forms of the VGGNet, were trained on the popular ImageNet images dataset. The VGG-16 and VGG-19 CNN were pre-trained deep neural networks on the ImageNet for computer vision (image recognition) problems, having 16 weight layers and 19 weight layers, respectively. They were used as pre-trained models to help learn the distinguishing features in COVID-19 X-ray images with the aid of a transfer learning approach, thus, trained DTL models for the identification of COVID-19 from X-ray images.

As shown in the workflow in Figs. 1 and 2, respectively, to train the VGG-16 based DTL model and VGG-19 based DTL model for the detection of COVID-19, the VGG-16 CNN and VGG-19 CNN was used as pre-trained models and were fine-tuned for COVID-19 detection based on the principles of transfer learning. The weights of the lower layers of the network, which train very common characteristics from the pre-trained model, were used as feature extractors for the implementation of transfer learning with fine-tuning. Therefore, the pre-trained model’s lower layers weights were frozen and therefore not updated through the training process, thus not participating in the transfer-learning process. The higher layers of the pre-trained model were used for learning task-specific features from the COVID-19 images dataset. In this case, the higher layers of the pre-trained model were unfrozen, made trainable or fine-tuned in which the weights of the layers were updated. Consequently, the layers were allowed to participate in the transfer-learning process. Each of these models ends with the SoftMax layer, which produces the outputs. The weights for the VGG-16 and VGG-19 networks were pre-trained on ImageNet, and the Fully Connected (FC) layer head was removed. From there, new fully-connected layer heads were built comprising POOL => FC = SOFTMAX layers and attached on top of VGG-16 and VGG-19. The Convolutional weights of VGG-16 and VGG19 were then frozen in such a way that only the FC layer head was trained.

Figure 1 The Architectural Workflow of the Proposed Fine-tuned VGG-16 based Deep Transfer Learning Model for COVID-19 Detection.

Figure 2 The Architectural Workflow of the Proposed Fine-tuned VGG-19 based Deep Transfer Learning Model for COVID-19 Detection.

Both binary class scenario and three-class classification scenario were considered in the workflow, in which the DTL model determines the class of the chest X-ray images as either “COVID-19” category or “Normal” category in the binary class scenario or as either “COVID-19” category, “Viral-Pneumonia” category, or “Normal” category in the three-class classification scenario.

The flowchart of the experimental algorithm for the deep transfer learning models based on the VGG-16 and VGG-19 networks proposed in this paper is presented in Fig. 3.

Figure 3 The Flowchart of the experimental algorithm for the DTL models based on the VGG-16 and VGG-19 networks.

Experimentation and results

Four different experiments were performed to classify radiological X-ray images using Deep Transfer Learning approaches. Two of the experiments (Experiments A and B) include each of the trained two Deep Transfer Learning models (VGG-16 and VGG-19 based) on binary class X-ray image dataset (with COVID-19 and Normal classes). The other two experiments (Experiments C and D) include each of the trained two Deep Transfer Learning models (VGG-16 and VGG-19 based) on a multiclass X-ray image dataset (with COVID-19, ViralPneumonia, and Normal classes). For these experiments, out of a total of 3,900 images used, 2,925 images (75%) were used for training the models, while 975 images (25%) were used for validation and to perform hyper-parameter tuning. A different set of 470 images (containing 70 COVID-19 images, 200 ViralPneumonia, and 200 Normal) were used for testing to obtain an impartial evaluation of the final model. The code for this experiment can be found at Fayemiwo et al. (2021b).

The performances of the proposed models were obtained from the models’ generated confusion matrices, using standard metrics such as accuracy, specificity, precision, recall (sensitivity), F1-Score, Matthews Correlation Coefficient and Cohen’s Kappa statistics. Equations 1, 2 and 3 show the formulas for computing the three key metrics used in this article, namely Accuracy, Matthews Correlation Coefficient and Cohen’s Kappa statistics, respectively:

(1) Accuracy=TP+TNTP+TN+FP+FN

(2) MCC=TP+TN−FP+FN(TP+FP)(TP+FN)(TN+FP)(TN+FN)

(3) Kappa=(TN+TP/TN+TP+FP+FN)−((TN+FP)×(TN+FN)/TN+TP+FP+FN)1−((TN+FP)×(TN+FN)/TN+TP+FP+FN)

where TP, TN, FP and FN denote True Positive, True Negative, False Positive and False Negative, respectively.

Experiment A

The first experiment was performed on the binary class dataset. A DTL model based on a pre-trained VGG-16 model was trained to classify the X-ray images into the two classes of COVID-19 or Normal; and to detect if X-ray images are simply of the class COVID-19 or Normal. The VGG-16 based DTL model summary detailing the layers and parameters in each layer of the model is shown in Table 1. The fine-tuned VGG-16 based DTL model consists of 14,747,650 total parameters, with 32,962 of them made trainable while 14,714,688 were non-trainable. The VGG-16 DTL model was modelled by employing a batch size of 10 in 40 epochs, using Adam optimizer specifically for updates of weights, cross-entropy loss function with a learning rate of 1e−2. Performance of the proposed fine-tuned VGG-16 based DTL model was evaluated on 25% of the X-ray images.

Table 1 The confusion matrix of the binary classification task obtained from the fine-tuned VGG-16 based DTL.

	Layers	Layer’s type	Shape of output	Num of trainable param	
1	Convolution_1 of Block_1	Convolution 2D	[64, 224, 224]	1,792	
2	Convolution_2 of Block_1	Convolution 2D	[64, 224, 224]	36,928	
3	Convolution_1 of Block_2	Convolution 2D	[128, 112, 112]	73,856	
4	Convolution_2of Block_2	Convolution 2D	[128, 112, 112]	147,584	
5	Convolution_1 of Block_3	Convolution 2D	[256, 56, 56]	295,168	
6	Convolution_2of Block_3	Convolution 2D	[256, 56, 56]	590,080	
7	Convolution_3of Block_3	Convolution 2D	[256, 56, 56]	590,080	
8	Convolution_1 of Block_4	Convolution 2D	[512, 28, 28]	1,180,160	
9	Convolution_2of Block_4	Convolution 2D	[512, 28, 28]	2,359,808	
10	Convolution_3of Block_4	Convolution 2D	[512, 28, 28]	2,359,808	
11	Convolution_1 of Block_5	Convolution 2D	[512, 14, 14]	2,359,808	
12	Convolution_2of Block_5	Convolution 2D	[512, 14, 14]	2,359,808	
13	Convolution_3of Block_5	Convolution 2D	[512, 14, 14]	2,359,808	
14	Flatten	Flatten	[512]	0	
15	Dense	Dense	[64]	32,832	
16	Dense_1	Dense	[3]	130	

The output of the confusion matrix for the binary classification as obtained from the VGG-16 based DTL model is shown in Table 2. Figure 4 illustrates the training loss and accuracy, the validation loss and accuracy graphs of the proposed fine-tuned VGG-16 based DTL model. The validation accuracy, recall, specificity, precision, F1-Score, Matthews Correlation Coefficient and Cohen’s Kappa statistics of the proposed fine-tuned VGG-16 based DTL model were also obtained.

Figure 4 The training loss and accuracy with the validation loss and accuracy curves obtained for the fine-tuned VGG-16 based Deep Transfer Learning Model (For Binary Classification).

Table 2 The precision, recall and F1-score obtained for the classification task using the fine-tuned VGG-16 based DTL model (binary classification).

	PREDICTED	
ACTUAL	COVID-19	NORMAL	
COVID-19	TP = 321	FN = 0	
NORMAL	FP = 5	TN = 324	

It was observed from Fig. 4 that the validation and training losses were slightly high in the earlier epochs and then almost flattened as the training occurs in subsequent epochs. The decrease in the loss values at around the 40th epoch was attributed to the fact that the fine-tuned VGG-16 based DTL model was exposed to all the available X-ray images time and again during each of the epochs considered during training.

The validation accuracy obtained for the fine-tuned VGG-16 based DTL model was 99.23%, its recall was 100%, while its specificity stands at 98.48%. The obtained values for the precision, recall, F1-Score, Matthews Correlation Coefficient and Cohen’s Kappa statistics metrics for the binary classification task using the VGG-16 based DTL model are given in Table 3.

Table 3 The confusion matrix of the binary classification task obtained from the fine-tuned VGG-19 based DTL.

	Precision	Recall	F1-score	MCC	Cohen Kappa	
COVID-19	0.98	1.00	0.99	0.98	0.98	
NORMAL	1.00	0.98	0.99	

Experiment B

The second experiment was performed on the binary class dataset. A DTL model based on a pre-trained VGG-19 model was trained to classify the X-ray images into the two classes of COVID-19 or Normal; and also, to detect if X-ray images are simply of the class COVID-19 or Normal. Also, the VGG-19 based DTL model summary detailing the layers and the parameters in each layer of the model is shown in Table 4. The fine-tuned VGG-19 based DTL model consists of 20,057,346 total parameters, with 32,962 of them made trainable while 20,024,384 were non-trainable. The VGG-19 DTL model was modelled by employing a batch size of 10 in 40 epochs, using Adam optimizer for the updates of weights, categorical cross-entropy loss function with a learning rate of 1e−1. Performance of the proposed fine-tuned VGG-19 based DTL model was evaluated on 25% of the X-ray images.

Table 4 The precision, recall and F1-score obtained for the classification task using the fine-tuned VGG-19 based DTL model (binary classification).

	Layers	Layer’s type	Shape of output	Num of trainable param	
1	Convolution_1 of Block_1	Convolution 2D	[64,224, 224]	1,792	
2	Convolution_2 of Block_1	Convolution 2D	[64,224, 224]	36,928	
3	Convolution_1 of Block_2	Convolution 2D	[128, 112, 112]	73,856	
4	Convolution_2 of Block_2	Convolution 2D	[128, 112, 112]	147,584	
5	Convolution_1 of Block_3	Convolution 2D	[256, 56, 56]	295,168	
6	Convolution_2 of Block_3	Convolution 2D	[256, 56, 56]	590,080	
7	Convolution_3 of Block_3	Convolution 2D	[256, 56, 56]	590,080	
8	Convolution_4 of Block_3	Convolution 2D	[256, 56, 56]	590,080	
9	Convolution_1 of Block_4	Convolution 2D	[512, 28, 28]	1,180,160	
10	Convolution_2 of Block_4	Convolution 2D	[512, 28, 28]	2,359,808	
11	Convolution_3 of Block_4	Convolution 2D	[512, 28, 28]	2,359,808	
12	Convolution_4 of Block_4	Convolution 2D	[512, 28, 28]	2,359,808	
13	Convolution_1 of Block_5	Convolution 2D	[512, 14, 14]	2,359,808	
14	Convolution_2 of Block_5	Convolution 2D	[512, 14, 14]	2,359,808	
15	Convolution_3 of Block_5	Convolution 2D	[512, 14, 14]	2,359,808	
16	Convolution_4 of Block_5	Convolution 2D	[512, 14, 14]	2,359,808	
17	Flatten	Flatten	[512]	0	
18	Dense	Dense	[64]	32,832	
19	Dense_1	Dense	[3]	130	

The output of the confusion matrix for the binary classification as obtained from the fine-tuned VGG-19 based DTL model are shown in Table 5. Figure 5 illustrates the training loss and accuracy along with the validation loss and accuracy graphs of the proposed fine-tuned VGG-19 based DTL model. The validation accuracy, recall, specificity, precision, and F1-Score of the proposed fine-tuned VGG-19 based DTL model were also obtained.

Figure 5 The training loss and accuracy with the validation loss and accuracy curves obtained for the fine-tuned VGG-19 based Deep Transfer Learning Model (For Binary Classification).

Table 5 The confusion matrix of the three-class classification task obtained from the fine-tuned VGG-16 based DTL.

	PREDICTED	
ACTUAL		COVID-19	NORMAL	
COVID-19	TP = 308	FN = 13	
NORMAL	FP = 0	TN = 329	

It was observed from Fig. 5 that the validation and training losses were slightly moderate in the earlier epochs and then decreases as the training occurs in more subsequent epochs (with a sharp increase at about epochs 12 and 20). The decrease in the loss values at around the 40th epoch were attributed to the fact that the fine-tuned VGG-19 based DTL model was exposed to all the available X-ray images repeatedly on all the epochs considered during training.

The validation accuracy obtained for the fine-tuned VGG-19 based DTL model was 98.00%, its recall was 95.95%, while its specificity stands at 100%. The values obtained for the precision, recall, F1-Score, Matthews Correlation Coefficient and Cohen’s Kappa statistics metrics for the binary classification task using the VGG-19 based DTL model are given in Table 6.

Table 6 The precision, recall and F1-Score obtained for the classification task using the fine-tuned VGG-16 based DTL model (three-class classification).

	Precision	Recall	F1-score	MCC	Cohen Kappa	
COVID-19	1.00	0.96	0.98	0.96	0.96	
NORMAL	0.96	1.00	0.98	

Experiment C

The third experiment was performed on the multiclass (three-class) dataset, in which a DTL model based on a pre-trained VGG-16 model was trained to classify the X-ray images into the three classes of COVID-19, Viral Pneumonia or Normal; and to detect if X-ray images were simply of the class COVID-19 or Viral Pneumonia or Normal. The VGG-16 based DTL model summary, detailing the layers and parameters in each layer of the model, is shown in Table 7. The fine-tuned VGG-16 based DTL model consists of 14,747,715 total parameters, with 33,027 of them made trainable while 14,714,688 were non-trainable. The VGG-16 DTL model was modelled by employing a batch size of 10 in 40 epochs, using Adam optimizer specifically for updates of weights, cross-entropy loss function with a learning rate of 1e−2, the performance of the proposed fine-tuned VGG-16 based DTL model was evaluated on 25% of the X-ray images.

Table 7 The confusion matrix of the three-class classification task obtained from the fine-tuned VGG-19 based DTL.

	Layers	Layer’s type	Shape of output	Num of trainable param	
1	Convolution_1 of Block_1	Convolution 2D	[64, 224, 224]	1,792	
2	Convolution_2 of Block_1	Convolution 2D	[64, 224, 224]	36,928	
3	Convolution_1 of Block_2	Convolution 2D	[128, 112, 112]	73,856	
4	Convolution_2 of Block_2	Convolution 2D	[128, 112, 112]	147,584	
5	Convolution_1 of Block_3	Convolution 2D	[256, 56, 56]	295,168	
6	Convolution_2 of Block_3	Convolution 2D	[256, 56, 56]	590,080	
7	Convolution_3 of Block_3	Convolution 2D	[256, 56, 56]	590,080	
8	Convolution_1 of Block_4	Convolution 2D	[512, 28, 28]	1,180,160	
9	Convolution_2 of Block_4	Convolution 2D	[512, 28, 28]	2,359,808	
10	Convolution_3 of Block_4	Convolution 2D	[512, 28, 28]	2,359,808	
11	Convolution_1 of Block_5	Convolution 2D	[512, 14, 14]	2,359,808	
12	Convolution_2 of Block_5	Convolution 2D	[512, 14, 14]	2,359,808	
13	Convolution_3 of Block_5	Convolution 2D	[512, 14, 14]	2,359,808	
14	Flatten	Flatten	[512]	0	
15	Dense	Dense	[64]	32,832	
16	Dense_1	Dense	[3]	195	

The output of the confusion matrix for the binary classification as obtained from the VGG-16 based DTL model are shown in Table 8. Figure 6 illustrates the training loss and accuracy and the validation loss and accuracy graphs of the proposed fine-tuned VGG-16 based DTL model. The validation accuracy, recall, specificity, precision, and F1-Score of the proposed fine-tuned VGG-16 based DTL model were also obtained.

Figure 6 The training loss and accuracy with the validation loss and accuracy curves obtained for the fine-tuned VGG-16 based Deep Transfer Learning Model (For Three-class Classification).

Table 8 The Precision, Recall and F1-Score obtained for the classification task using the fine-tuned VGG-19 based DTL model (Three-class Classification).

	PREDICTED	
ACTUAL	COVID-19	NORMAL	ViralPneumonia	TOTAL	
COVID-19	339	1	6	346	
NORMAL	0	321	18	339	
ViralPneumonia	4	31	255	290	
TOTAL	343	353	279	975	

It was observed from Fig. 6 that the validation and training losses were significantly high in the earlier epochs and then slowly decrease as the training occurs in more subsequent epochs. The decrease in the loss values at around the 40th epoch was attributed to the fact that the fine-tuned VGG-16 based DTL model was exposed to all the available X-ray images repeatedly during each of the epochs considered during training.

The validation accuracy obtained for the fine-tuned VGG-16 based DTL model was 93.85%, its recall was 97.98%, while its specificity stands at 94.69%. The obtained values for the precision, recall, F1-Score, Matthews Correlation Coefficient and Cohen’s Kappa statistics metrics for the three-class classification task using the VGG-16 based DTL model are given in Table 9.

Table 9 Comparison of the proposed COVID-19 diagnostic methods with other deep learning methods developed using radiology images.

	Precision	Recall	F1-score	MCC	Cohen Kappa	
COVID-19	0.99	0.98	0.98	0.91	0.91	
NORMAL	0.91	0.95	0.93	
ViralPneumonia	0.91	0.88	0.90	

Experiment D

The fourth experiment performed on the multiclass (three-class) dataset, in which a DTL model based on a pre-trained VGG-19 model was trained to classify the X-ray images into the three classes of COVID-19, ViralPneumonia or Normal; and also, to detect if X-ray images are simply of the class COVID-19 or ViralPneumonia or Normal. The VGG-19 based DTL model summary detailing the layers and parameters in each layer of the model is shown in Table 10. The fine-tuned VGG-19 based DTL model consists of 20,057,411 total parameters, with 33,027 of them made trainable while 20,024,384 were non-trainable. The VGG-19 DTL model was modelled by employing a batch size of 10 in 40 epochs, using Adam optimizer specifically for updates of weights, cross-entropy loss function with a learning rate of 1e−1, the performance of the proposed fine-tuned VGG-19 based DTL model was evaluated on the 25% of the X-ray images.

Table 10 The layers and layer parameters of the proposed fine-tuned VGG-19 based DTL model (Three-class Classification).

	Layers	Layer’s type	Shape of output	Num of trainable param	
1	Convolution_1 of Block_1	Convolution 2D	[64, 224, 224]	1,792	
2	Convolution_2 of Block_1	Convolution 2D	[64, 224, 224]	36,928	
3	Convolution_1 of Block_2	Convolution 2D	[128, 112, 112]	73,856	
4	Convolution_2 of Block_2	Convolution 2D	[128, 112, 112]	147,584	
5	Convolution_1 of Block_3	Convolution 2D	[256, 56, 56]	295,168	
6	Convolution_2 of Block_3	Convolution 2D	[256, 56, 56]	590,080	
7	Convolution_3 of Block_3	Convolution 2D	[256, 56, 56]	590,080	
8	Convolution_1 of Block_4	Convolution 2D	[512, 28, 28]	1,180,160	
9	Convolution_2 of Block_4	Convolution 2D	[512, 28, 28]	2,359,808	
10	Convolution_3 of Block_4	Convolution 2D	[512, 28, 28]	2,359,808	
11	Convolution_1 of Block_5	Convolution 2D	[512, 14, 14]	2,359,808	
12	Convolution_2 of Block_5	Convolution 2D	[512, 14, 14]	2,359,808	
13	Convolution_3 of Block_5	Convolution 2D	[512, 14, 14]	2,359,808	
14	Flatten	Flatten	[512]	0	
15	Dense	Dense	[64]	32,832	
16	Dense_1	Dense	[3]	195	

The output of the confusion matrix for the binary classification as obtained from the VGG-19 based DTL model are shown in Table 11. Figure 7 illustrates the training loss and accuracy and the validation loss and accuracy graphs of the proposed fine-tuned VGG-16 based DTL model. The validation accuracy, recall, specificity, precision, and F1-Score of the proposed fine-tuned VGG-16 based DTL model were also obtained.

Figure 7 The training loss and accuracy with the validation loss and accuracy curves obtained for the fine-tuned VGG-19 based Deep Transfer Learning Model (For Three-class Classification).

Table 11 The confusion matrix of the three-class classification task obtained from the fine-tuned VGG-19 based DTL.

	PREDICTED	
ACTUAL	COVID-19	NORMAL	ViralPneumonia	TOTAL	
COVID-19	332	1	13	346	
NORMAL	2	304	33	339	
ViralPneumonia	1	19	270	290	
TOTAL	335	324	316	975	

It was observed from Fig. 7 that the validation and training losses were significantly high in the earlier epochs and then gradually decrease as the training occurs in more subsequent epochs. This decrease in the loss values at around the 40th epoch was attributed to the fact that the fine-tuned VGG-19 based DTL model was exposed to all the available X-ray images time and again during each of the epochs considered during training.

The validation accuracy obtained for the fine-tuned VGG-19 based DTL model was 92.92%, its recall was 95.95%, while its specificity stands at 89.68%. The obtained values for the precision, recall, F1-Score, Matthews Correlation Coefficient and Cohen’s Kappa statistics metrics for the three-class classification task using the VGG-19 based DTL model are given in Table 12.

Table 12 The precision, recall, F1-score, Matthews correlation coefficient and Cohen’s Kappa statistics obtained for the classification task using the fine-tuned VGG-19 based DTL model (three-class Classification).

	Precision	Recall	F1-score	MCC	Cohen Kappa	
COVID-19	0.99	0.96	0.98	0.89	0.89	
NORMAL	0.94	0.90	0.92	
ViralPneumonia	0.85	0.93	0.89	

It was noted from the obtained confusion matrices and the computed performance evaluation metrics of the binary and three-class classification tasks that the fine-tuned VGG-16 based deep transfer learning model outperformed the fine-tuned VGG-19 based deep transfer learning model in the detection of COVID-19. Based on this, some tests were carried out on unlabeled images using the developed fine-tuned VGG-16 multi-classification model. The test was carried out to obtained an impartial evaluation of the final model. Some results of the tests are shown in Figs. 8 to 10.

Figure 8 COVID-19 sample test results with the predicted level of confidence value.

Figure 10 Normal sample test results with the predicted level of confidence value.

Being the best performing model in this study, fine-tuned VGG-16 DTL model was tested on the test dataset of 470 images. The test accuracy obtained for the model was 98%. The results of the tests as shown in Figs. 8 to 10 show how the fine-tuned VGG-16 DTL model classified and detected each of the images as either “COVID-19”, “Viral Pneumonia”, or “Normal.” The level of confidence in the model classification is also shown. Figure 8 shows the sample images that were detected as “COVID-19” along with the model’s classification confidence accuracy values. Figure 9 shows the sample images that were detected as “Viral Pneumonia” and the model’s classification confidence accuracy values. In contrast, Fig. 10 shows the sample images that were detected as “Normal” along with the model’s classification confidence accuracy values.

Figure 9 Viral Pneumonia sample test results with a predicted level of confidence value.

Out of the six sample images shown in Fig. 8, only one showed a lower confidence level of 76.37%, while others were above 94%. Similar results could be seen in Fig. 10 for Normal classification, where the lowest confidence level is 78.92%. However, the lowest output for Viral Pneumonia is 96.21%, as shown in Fig. 9. These test results showed that the developed models could generalize and adapt to new data outside the training and validation dataset. These test results are necessary to show the adaptability of the developed models when related data is considered.

Evaluation of results

The results obtained in this work were compared with thirteen other existing approaches in the literature. Few studies conducted before this study had used twenty-five and fifty images in each class (National Health Commission of People’s Republic of China, 2020; Hemdan, Shouman & Karar, 2020; Narin, Kaya & Pamuk, 2020), while nine out of the twelve approaches benchmarked used imbalanced data (Table 9). Generally, the problem in modeling imbalanced data is that it could lead to the inability of the model to generalize, or the model can be biased towards a class with a high number of data points. Hence, in this study, an equal value of data (1,300 images) was used for each category, and this is believed to have contributed to increasing accuracies of the proposed models. At the moment, creating an automated diagnostic tool for the detection of COVID-19 suffers from the drawback of limited number of cases. To ensure the generalization of the models developed in this work, data augmentation was performed by setting the random image rotation setting to 15 degrees clockwise. The proposed new models were based on fine-tuning VGG-16 and VGG-19 methods by constructing a new fully-connected layer head consisting of POOL => FC = SOFTMAX layers and append it on top of VGG-16 and VGG-19; the Convolution weights of VGG-16 and VGG-19 were then frozen, such that only the FC layer head was trained. The fine-tuned models gave better results than other models that used ordinary pre-trained VGG-16 and VGG-19 (Apostolopoulos & Mpesiana, 2020; El Asnaoui & Chawki, 2020). Complete results of the comparison with thirteen other existing results from the literature are presented in Table 13, with the proposed model recording the best performance accuracy. The closest performing model to the proposed model is that of the DarkCovidNet model (Ozturk et al., 2020) with 98.08% accuracy, while the proposed DTL-based VGG-16 model has 99.23% accuracy, both in the binary classification task.

Table 13 Comparison of the proposed COVID-19 diagnostic methods with other deep learning methods developed using radiology images.

S/no	Study	Type of images	Number of cases	Method used	Accuracy (%)	
1	Apostolopoulos & Mpesiana (2020)	Chest X-ray	224 COVID-19 (+)
700 Pneumonia
504 Healthy	VGG-19	93.48	
2	Wang & Wong (2020)	Chest X-ray	53 COVID-19 (+)
5526 COVID-19 (−)
8066 Healthy	COVID-Net	92.4	
3	National Health Commission of People’s Republic of China (2020)	Chest X-ray	25 COVID-19 (+)
25 COVID-19 (−)	ResNet50+ SVM	95.38	
4	Hemdan, Shouman & Karar (2020)	Chest X-ray	25 COVID-19 (+)
25 Normal	COVIDX-Net	90.0	
5	Narin, Kaya & Pamuk (2020)	Chest X-ray	50 COVID-19 (+)
50 COVID-19 (−)	Deep CNN
ResNet-50	98.0	
6	Liu et al., (2020)	Chest CT	777 COVID-19 (+)
708 Healthy	DRE-Net	86.0	
7	Wang et al. (2020b)	Chest CT	195 COVID-19 (+)
258 COVID-19 (−)	M-Inception	82.9	
8	Zheng et al. (2020)	Chest CT	313 COVID-19 (+)
229 COVID-19 (−)	UNet+3D Deep
Network	90.8	
9	Xu et al. (2020)	Chest CT	219 COVID-19 (−)
224 Viral pneumonia
75 Healthy	ResNet + Location
Attention	86.7	
10	El Asnaoui & Chawki (2020)	Chest X-ray and CT Scan	2,780 bacterial pneumonia
1,493 corona-virus
231 Covid19
1,583 normal	Inception_Resnet_V2	92.18	
DensNet201	88.09	
Resnet50	87.54	
Mobilenet_V2	85.47	
Inception_V3	88.03	
VGG-16	74.84	
VGG-19	72.52	
11	Ozturk et al. (2020)	Chest X-ray	2-Class:
125 COVID-19 (+)
500 No-Findings	DarkCovidNet	98.08	
3-Class:
125 COVID-19 (+)
500 Pneumonia
500 No-Findings	DarkCovidNet	87.02	
12	Elzeki et al. (2021)	Chest X-ray	2-Class:
221 COVID-19 (+)
234 Normal	CXRVN Network	96.7	
3-Class:
221 COVID-19 (+)
148 Pneumonia
234 Normal	CXRVN Network	93.07	
13	Irfan et al. (2021)	Chest X-ray and Chest CT	3,500 COVID-19 and 1,500 healthy	CNN and Long-short term memory	99%	
14	Proposed Study	Chest X-ray	2-Class:
1,300 COVID-19 (+)
1,300 Normal	VGG-16	99.23	
VGG-19	98.00	
3-Class:
1,300 COVID-19 (+)
1,300 Viral Pneumonia
1,300 Normal	VGG-16	93.85	
VGG-19	92.92	

Conclusions and future work

Several researchers around the world are combining their efforts to collect data and develop solutions for the COVID-19 pandemic problem. Laboratory testing of suspected cases characterized by long waiting periods and an exponential increase in demand for tests has hitherto constituted a significant bottleneck globally. Hence, rapid diagnostic test kits are being developed, most of which are currently undergoing clinical validation and have yet to be adopted for routine use. This paper proposed a solution using the Deep Learning Convolutional Neural Network Model to classify a real-life COVID-19 dataset of chest X-ray images into three-classes: COVID-19, Viral-Pneumonia and Normal categories. Two experiments were performed where the VGG-16 and VGG-19 CNN with DTL was implemented in Jupyter Notebook using Python programming language. Experimental results showed that the pre-trained VGG-16 DTL model classified COVID-19 data better than the VGG-19 based DTL model. The fine-tuned VGG-16 and VGG-19 models produced classification accuracies of 99.23% and 98.00%, respectively, for binary classification and 93.85% and 92.92% for multiclass classification. The proposed model, therefore, outperformed existing methods in terms of accuracy. Moreover, the fine-tuned VGG-16 and VGG-19 models have MCC of 0.98 and 0.96 respectively in the binary classification, and 0.91 and 0.89 for multiclass classification. These results showed that there are strong positive correlations between the models’ predictions and the true labels. In the two classification tasks (binary and three-class), it was observed that the fine-tuned VGG-16 DTL model had stronger positive correlations in the MCC metric than the fine-tuned VGG-19 DTL model. The VGG-16 DTL model has a Kappa value of 0.98 as against 0.96 for the VGG-19 DTL model in the binary classification task, while in the three-class classification problem, the VGG-16 DTL model has a Kappa value of 0.91 as against 0.89 for the VGG-19 DTL model. This result is in agreement with the trend observed in the MCC metric. The findings of this study have a high potential of increasing the prediction accuracy for COVID-19 disease, which would be of immense benefit to the medical field and the entire human populace as it could help save many lives from untimely death.

The researchers suspect that the better performance of the VGG-16 DTL model might be attributed to the volume of data used in the experiments; that is, the depth of layers in the VGG-19 architecture may not have any significant effect on the performance when the dataset is small. This suspicion would be investigated in future work when more COVID-19 data is available. Finally, the COVID-19 images from Chest CT scans are not readily available, unlike X-ray images, because of their high cost. Therefore, other future works would consider using Chest CT images to develop a more sensitive diagnostic tool for detecting viral pneumonia and COVID-19 variants. Further hyper-parameter tweaking would also be done to get more accurate results.

The authors acknowledge the efforts of Dr. (Mrs.) F. O. Unuabonah in proofreading this article and increasing its readability.

Additional Information and Declarations

Competing Interests

Author Contributions

Data Availability

The authors declare that they have no competing interests.

Michael Adebisi Fayemiwo conceived and designed the experiments, performed the experiments, analyzed the data, performed the computation work, prepared figures and/or tables, authored or reviewed drafts of the paper, and approved the final draft.

Toluwase Ayobami Olowookere conceived and designed the experiments, performed the experiments, analyzed the data, performed the computation work, prepared figures and/or tables, authored or reviewed drafts of the paper, and approved the final draft.

Samson Afolabi Arekete conceived and designed the experiments, performed the experiments, analyzed the data, performed the computation work, prepared figures and/or tables, authored or reviewed drafts of the paper, and approved the final draft.

Adewale Opeoluwa Ogunde conceived and designed the experiments, performed the experiments, analyzed the data, performed the computation work, prepared figures and/or tables, authored or reviewed drafts of the paper, and approved the final draft.

Mba Obasi Odim conceived and designed the experiments, performed the experiments, analyzed the data, performed the computation work, prepared figures and/or tables, authored or reviewed drafts of the paper, and approved the final draft.

Bosede Oyenike Oguntunde conceived and designed the experiments, performed the experiments, analyzed the data, performed the computation work, prepared figures and/or tables, authored or reviewed drafts of the paper, and approved the final draft.

Oluwabunmi Omobolanle Olaniyan conceived and designed the experiments, performed the experiments, analyzed the data, performed the computation work, prepared figures and/or tables, authored or reviewed drafts of the paper, and approved the final draft.

Theresa Omolayo Ojewumi conceived and designed the experiments, performed the experiments, analyzed the data, performed the computation work, prepared figures and/or tables, authored or reviewed drafts of the paper, and approved the final draft.

Idowu Sunday Oyetade conceived and designed the experiments, performed the experiments, analyzed the data, performed the computation work, prepared figures and/or tables, authored or reviewed drafts of the paper, and approved the final draft.

Ademola Adegoke Aremu conceived and designed the experiments, performed the experiments, analyzed the data, performed the computation work, prepared figures and/or tables, authored or reviewed drafts of the paper, and approved the final draft.

Aderonke Anthonia Kayode conceived and designed the experiments, performed the experiments, analyzed the data, performed the computation work, prepared figures and/or tables, authored or reviewed drafts of the paper, and approved the final draft.

The following information was supplied regarding data availability:

The data is available at figshare: Fayemiwo, Michael; Olowookere, Toluwase; Arekete, Samson; Ogunde, Adewale; Odim, Mba; Oguntunde, Bosede; et al. (2021): RAIG COVID19 compiled dataset.zip. figshare. Dataset. DOI 10.6084/m9.figshare.14151854.v4.

The code is also available at figshare: Fayemiwo, Michael; Olowookere, Toluwase; Arekete, Samson; Ogunde, Adewale; Odim, Mba; Oguntunde, Bosede; et al. (2021): RAIG: classification of COVID-19 Radiology Dataset. figshare. Software. DOI 10.6084/m9.figshare.14151971.v3.

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
