# Peer review of "Modeling a deep transfer learning framework for the classification of COVID-19 radiology dataset"

_PeerJ Computer Science, doi:10.7717/peerj-cs.614_

## Round 0.1 · original submission · Major Revisions

The article presents a study based on deep transfer learning to detect COVID among X-rays images of patients.

As the reviewers mentioned, the article is interesting, but some improvements are needed.

In particular, as one of the reviewers points out, some details should be incorporated (batch size, reason for the images to be downsized, number of training epochs).

Here are some other points to address.

Major points:

1- The binary classification results generated with confusion matrices must report the Matthews correlation coefficient (MCC) values as well. The discussion of the results should mention and comment the MCC's obtained by the classifiers.

2- The abstract should report the results measured with the MCC, and not with the accuracy.

3- The resolution of Figure 1 must be improved.

4- In Tables 2, 4, 6, and 8, the authors should add more statistical rates such as the Matthews correlation coefficient (MCC), true negative rate, negative predictive value, accuracy, and Cohen's kappa.

5- The abstract should not report the learning rate values of the artificial neural networks.

Minor points:

6- The authors sometimes write "X-rays" and sometimes write "x-rays". Please unify.

7- The URL of the dataset on FigShare on line 425 should be replaced with a reference to the same link.

Please also study and address the points raised by the three reviewers.

Reviewer 1 ·

Basic reporting

The Primary contribution of the paper looks interesting and presentation also looks fine. Literature review aspect is also very nice. And results and overall article structure is also fine. All the terminologies are used promptly.
The Novelty of the paper is also good and even the paper can be used for real-time analysis.

Experimental design

The Methodology is also good and nice work contribution is observed in the paper.
Research questions are well defined. But Research Gaps needs more work to be done in proper form.

Validity of the findings

The results are Ok. And conclusions and analysis is written and described in proper manner.
Data is also good. Conclusions are well stated and even the problem is well laid and defined in the paper.

Additional comments

The Paper needs the following Technical Revisions and is Requested for re-review:

1. Add Objectives of the paper at the end of Introduction. Add Organization of the paper.
2. No need for any sub-heading "RELATED WORKS". Just take one heading "Literature review" and merge everything there. At the end of Literature review, highlight in 9-15 lines what overall technical gaps are observed in existing works, that led to the design of the proposed methodology.
3. Add the Methodology aspect to this paper- Your proposed model, Algorithm or Flowchart of the Data Analysis.
4. Add more Analysis to this aspect.
5. Add future scope to this paper.
6. Add the following references to this paper:

1. Zivkovic, M., Bacanin, N., Venkatachalam, K., Nayyar, A., Djordjevic, A., Strumberger, I., & Al-Turjman, F. (2021). COVID-19 cases prediction by using hybrid machine learning and beetle antennae search approach. Sustainable Cities and Society, 66, 102669.
2. Kumar, A., Sharma, K., Singh, H., Srikanth, P., Krishnamurthi, R., & Nayyar, A. Drone-Based Social Distancing, Sanitization, Inspection, Monitoring, and Control Room for COVID-19. Artificial Intelligence and Machine Learning for COVID-19, 153.
3. Devi, A., & Nayyar, A. Perspectives on the Definition of Data Visualization: A Mapping Study and Discussion on Coronavirus (COVID-19) Dataset. Emerging Technologies for Battling Covid-19: Applications and Innovations, 223.
4. Sharma, K., Singh, H., Sharma, D. K., Kumar, A., Nayyar, A., & Krishnamurthi, R. Dynamic Models and Control Techniques for Drone Delivery of Medications and Other Healthcare Items in COVID-19. Emerging Technologies for Battling Covid-19: Applications and Innovations, 1.
5. Alzubi, J., Nayyar, A., & Kumar, A. (2018, November). Machine learning from theory to algorithms: an overview. In Journal of physics: conference series (Vol. 1142, No. 1, p. 012012). IOP Publishing.

Reviewer 2 ·

Basic reporting

-

Experimental design

-

Validity of the findings

-

Additional comments

-please add colorful picture of measurements (optionally);;; + arrows what is what;;;;
-please add block diagram of the proposed research;;;;
-please add photo/photos of application of the proposed research ;;;; what is result of the analysis? ;;;;;
-please add sentences about future analysis;;;
-Figures should have better quality;;;;
-please add arrows to photos what is what;;;
-formulas and fonts should be formatted;;;;
-fonts in figures should be bigger;;;;
-please add labels to figures;;;
-references should be 2018-2021 Web of Science about 50% or more ;; 30-40 at least.;;;;
-Please compare with other methods, justify. Advantages or Disadvantages of neural network;;;
-is there possibility to use the proposed methods for other problems? ;;;;
for example;;;;

1)
Irfan, M.; Iftikhar, M.A.; Yasin, S.; Draz, U.; Ali, T.; Hussain, S.;
Bukhari, S.; Alwadie, A.S.; Rahman, S.; Glowacz, A.; et al.
Role of Hybrid Deep Neural Networks (HDNNs), Computed Tomography,
and Chest X-Rays for the Detection of COVID-19.
Int. J. Environ. Res. Public Health
2021, 18, 3056.
https://doi.org/10.3390/ijerph18063056

-Conclusion: point out what are you done;;;;

Reviewer 3 ·

Basic reporting

The title, abstract, introduction, methods, results and discussion are appropriate for the content of the text. Furthermore, the article is well constructed, the experiments are well conducted, and analysis is well performed. The figures are relevant, but the quality of the figures are low. The figures are not labelled and described well.

Experimental design

The angle of the research is original and the research is within the scope of the journal. Research question is well defined, relevant and meaningful. The overview and their proposal for a more suitable technology is highly technical, ethical and logistical.

Validity of the findings

The introduction is comprehensive. The findings are meaningful. The conclusions are well stated and relevant to the research questions.

Additional comments

This study trained a Deep Transfer-Learning Model (DTL) to classify the COVID-19 radiology dataset of chest X-ray images in both binary (COVID-19 or Normal) and three-class (COVID-19, Viral-Pneumonia or Normal) classification scenarios. The authors fine-tuned the model further for optimal performance. In their experiments, they achieved a test accuracy of 98%.



Editorial Criteria
BASIC REPORTING
The title, abstract, introduction, methods, results and discussion are appropriate for the content of the text. Furthermore, the article is well constructed, the experiments are well conducted, and analysis is well performed. The figures are relevant, but the quality of the figures are low. The figures are not labelled and described well.
EXPERIMENTAL DESIGN
The angle of the research is original and the research is within the scope of the journal. Research question is well defined, relevant and meaningful. The overview and their proposal for a more suitable technology is highly technical, ethical and logistical.
VALIDITY OF THE FINDINGS
The introduction is comprehensive. The findings are meaningful. The conclusions are well stated and relevant to the research questions.

Overall, I think this study is novel and will be of interest to others in the community of Artificial Intelligence Applications for COVID-19. This study proposed a solution by using Deep Learning Convolutional Neural Network Model to classify the COVID-19 X-ray images into three classes: COVID-19, Viral-Pneumonia or Normal categories. This proposed model also outperformed existing methods in terms of accuracy. Some of the weaknesses are the not always easy readability of the text which establishes unclear logical links between concepts. In general, the work is convincing except some major and minor comments below:


Major Comments:

The authors included multiple datasets (Qatar University, University of Dhaka and Mendeley dataset). I wonder if the batch size and the number of training epochs vary from one dataset to another. Are they application dependent?

I noticed that the X-ray images were resized to 224×224 pixels? How was the size decided? I wonder if the accuracies on different image sizes were computed?



Minor Comments:
Figure 4, 5, 6, 7: recommend improving the resolution; recommend removing the background grey color and grid lines.

Figure 8, 9, 10: the red/green.blue texts are really hard to read. Please make them clear. Please remove the “Output” on top of each subplot.

Line 670-671, “The fine-tuned models gave better results when also compared to other models that used ordinary pre-trained VGG-16 and VGG-19”: this sentence needs to remove “when also”.

A reference is needed for OpenCV in line 427.

It will be more clear if the authors can add the equations for the dimension and number of parameters calculation.

It will be more clear to have a section of Abbreviations.

Annotated reviews are not available for download in order to protect the identity of reviewers who chose to remain anonymous.

---

## Round 0.2 · accepted · Accept

The authors correctly addressed the points raised by the reviewers and made the proper edits to the article to make it suitable for publication, as the reviewer noted. I suggest the article can be accepted for publication in PeerJ Computer Science.

Reviewer 1 ·

Basic reporting

The Revised paper looks Ok and can be recommended for Publication. Now the literature, technical flow and figures looks Ok.

Experimental design

The experimental design looks broad and elaborate. And can be accepted.

Validity of the findings

The Results and ANALYSIS section is Ok and well elaborated.

Additional comments

The Paper stands Accepted with no further revisions.